# Science of music-based citizen science: How seeing influences hearing

**Daniel Bedoya**[1‡*], **Paul Lascabettes**[2‡], **Lawrence Fyfe**[3‡], **Elaine Chew**[3‡]

**1** Laboratoire de Mécanique des Structures et des Systèmes Couplés (LMSSC), Conservatoire National des Arts et Métiers (CNAM), Paris, France, **2** Institut de Recherche Mathématique Avancée IRMA, University of Strasbourg, Strasbourg, France, **3** Department of Engineering and School of Biomedical Engineering and Imaging Sciences, King's College London, London, United Kingdom

‡ All authors conducted this work while at the STMS Laboratoire (UMR9912) – CNRS, Sorbonne Université, IRCAM, Ministère de la Culture, Paris, France
* daniel.bedoya@ircam.fr

**Data availability statement:** The authors confirm that all data underlying the findings are available without restriction. All data not contained within the paper or supporting files (including raw data, code, video performances, and interactive figures) are available on figshare

## Abstract

Citizen science engages volunteers to contribute data to scientific projects, often through visual annotation tasks. Hearing based activities are rare and less well understood. Having high quality annotations of performed music structures is essential for reliable algorithmic analysis of recorded music with applications ranging from music information retrieval to music therapy. Music annotations typically begin with an aural input combined with a variety of visual representations, but the impact of the visuals and aural inputs on the annotations are not known. Here, we present a study where participants annotate music segmentation boundaries of variable strengths given only visuals (audio waveform or piano roll) or only audio or both visuals and audio simultaneously. Participants were presented with the set of 33 contrasting theme and variations extracted from a through-recorded performance of Beethoven's 32 Variations in C minor, WoO 80, under differing audiovisual conditions. Their segmentation boundaries were visualized using boundary credence profiles and compared using the unbalanced optimal transport distance, which tracks boundary weights and penalizes boundary removal, and compared to the F-measure. Compared to annotations derived from audio/visual (cross-modal) input (considered as the gold standard for our study), boundary annotations derived from visual (unimodal) input were closer than those derived from audio (unimodal) input. The presence of visuals led to larger peaks in boundary credence profiles, marking clearer global segmentations, while audio helped resolve discrepancies and capture subtle segmentation cues. We conclude that audio and visual inputs can be used as cognitive scaffolding to enhance results in large-scale citizen science annotation of music media and to support data analysis and interpretation. In summary, visuals provide cues for big structures, but complex structural nuances are better discerned by ear.

## Introduction

Citizen science presents an engaging scientific approach to harness volunteer thinking for crowdsourced annotations to assist in tackling difficult-to-solve computational problems

repositories at the following links: (https://doi.org/10.6084/m9.figshare.27918021), (https://doi.org/10.6084/m9.figshare.27917934), (https://doi.org/10.6084/m9.figshare.27917781), (https://doi.org/10.6084/m9.figshare.24635568.v1), and (https://doi.org/10.6084/m9.figshare.c.6755340.v1).

**Funding:** This research was part of the COSMOS project that has received funding from the European Research Council under the European Union's Horizon 2020 research and innovation program (Grant agreement No. 788960). Participant data was collected at the INSEAD-Sorbonne Université Behavioral Lab; where the protocol was approved by their ethics committee, and the experiments received funding from the French Excellence Initiative (Idex) at Sorbonne Université. Paul Lascabettes was supported by a Specific Doctoral Contract for Normaliens (CDSN). There was no additional external funding received for this study.

**Competing interests:** The authors have declared that no competing interests exist.

in domains ranging from astronomy [1] to ecology [2] to medical imaging [3]. Music is no exception. One of the most popular tasks for which humans possess excellent intuition is segmentation. The crowdsourcing of music segmentation has led to structural analysis of large music corpora [4]. Obtaining quality data is of the essence to best understand human perception of auditory structures in performed music.

When machines process music, the digitized music data can be loaded directly into algorithms or AI-based agents for the extracting of structural segmentation. In contrast, when humans listen to and annotate music, this is often a complex task where music is primarily auditory, but it can involve more than one sense. When one or more sensory inputs influence what is perceived by another, we speak about cross-modal or intersensory perception [5]. It may happen, for example, to an audience, a performer, or a data science annotator. Audiences seek the multiple sensations of live music performances and dance or even the enhanced aspects of music videos. Performers are constantly immersed in the multisensorial act of the physical gesture, auditory feedback, and even (sometimes) reading music notation, and following a conductor or a fellow player. There is evidence that in tasks involving auditory stimuli, annotators may experience natural cross-modal links between auditory and visual information (such as the sound of a rising pitch represented by notes moving up on a page), even when an activity is only focused on one modality [6]. Here, we compare annotations created with co-occurring or alternating auditory and visual support in a structural segmentation task, where participants mark boundaries using an interactive annotation platform.

Many types of visuals can have an influence in the perception of music structures, notably in the domains of music learning, appreciation, and understanding. First, many musicians learn to read and interpret music through notation. Even in a short eight-measure unit, the eye comprehends musical form better than the ear [7]. Second, concerning music appreciation, visual information prevails over aural when musicians and non-musicians judge the outcomes of music performance competitions [8], even though sound is the most important source of information for this task. The visual component in music perception is essential in communicating meaning, influencing an audience's evaluation of music performance [9]. For example, the physical gestures of a percussionist can alter note duration perception [10]. Third, several approaches use visualizations to promote a better understanding of music structures. For instance, Chew [11]'s spiral array uses points and triangles on nested helices to represent perceived distances between pitches, chords, and keys for real-time chord and key tracking, as demonstrated in 3D visualizations and in augmented reality by François [12]. Cruz et al. [13] proposed a visual language using geometrical shapes of different sizes and colors to identify pitch and timbre. Malandrino et al. [14] created software that maps chord changes in traditional four-part harmony to colored rectangles. In parallel, Miyazaki et al. [15] developed a system that uses 3-dimensional, multicolored explorations of MIDI data as a means to provide different perspectives of musical structures. To control the scope of this study and the influence of these vastly different visual components, our experiments are centered on waveform and piano roll visualizations, which are widely used in music information research [16].

To our knowledge, no study to date has compared the differences between boundary annotations in unimodal (aural or visual) vs. cross-modal (aural and visual) conditions. This study aims to explore that question using the web-based annotation platform CosmoNote [17] to annotate segmentation boundaries of four strength levels based on the concept of prosodic functions, namely, segmentation and prominence [18]. To assess how aggregated segmentation annotations differ when participants mainly use their eyes rather than their ears, we designed a study where participants were exposed to combinations of visual/aural cues and could freely place or remove boundaries. However, boundary annotations over a

time axis placed by humans are prone to subjectivity, ambiguity, and precision errors within and between annotators [19]. To better capture this temporal nuance when comparing two boundary annotation curves we used the unbalanced optimal transport distance, which has already been shown to work well with musical boundary credence profiles [20]. We use a bespoke unbalanced optimal transport algorithm to fit our analysis, allowing comparisons between a significantly larger number of boundary annotations and offering crucial insights into which boundaries determine the differences between two given curves. We contrast this technique to the widely used F-measure [21], arguing that our algorithm is well suited to closely analyze the difference between distributions representing weighted boundary annotations in performed music.

## Materials and methods

### Participants

The study was approved by the INSEAD Institutional Review Board (ID 202063) and was conducted between April 14th and 15th, 2022 at the INSEAD-Sorbonne Université Behavioural Lab. Participants signed an informed consent form where the goal, context, duration, tasks, compensation, and voluntary nature of the experiment were explained. The data were analyzed anonymously.

Boundary annotations were collected from 56 participants (31 female, 25 male). Participants were classified by age group: 58% between 18-24 years old, 40% between 25-34 years old, and 2% between 35-44 years old. The group was divided equally into musicians and non-musicians. Musicians reported at least five years of musical practice and one year of formal music theory training.

### Annotation interface

The study was conducted using the CosmoNote web-based annotation platform [17]. This tool allows participants to interactively visualize, listen to, and annotate music through a web browser. CosmoNote was used for this study because of three characteristics: its visualizations, annotation capabilities, and customizable settings.

A large horizontal pane presents visual representations synchronized to the audio signals that can be zoomed and panned. Multiple representations can be overlaid and toggled to preserve vertical space; see Fig 1. Two such visuals relevant to our study are: (1) the piano roll representation, which draws notes as horizontal rectangles filled with gradients of blue. It represents time horizontally, pitch vertically (aligned to a vertical piano keyboard on the left of the screen), and MIDI note velocity (correlated with loudness) through the transparency of the color fill, with more transparent notes being softer; and, (2) the waveform representation draws thin vertical green lines indicating the amplitude of the sound waves (vertically) over time (horizontally).

CosmoNote has four annotation types that can have text labels: (1) Boundaries, vertical markers with four strength levels, displayed as red vertical lines of different widths; see Fig 1; (2) Regions, ranged time selections; (3) Note groups, sets of one or many notes selected by users; and, (4) Comments, informative text markers represented by dashed vertical lines. Our study focused on segmentation by using only the boundary annotations.

CosmoNote's modular platform can change its features to create custom annotation experiments. Researchers can force the interface to show or hide any visuals or annotation types to suit a desired setup.

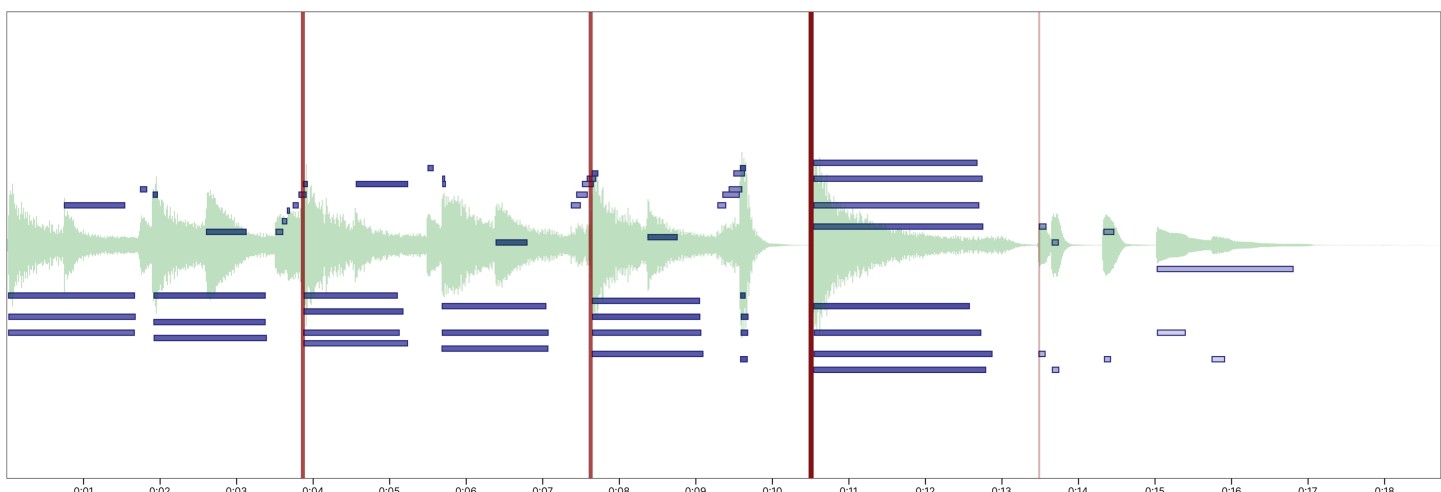

**Fig 1. CosmoNote's interface.** Visualization pane showing a piano roll (blue rectangles), a waveform (green curve), and boundaries with different levels (vertical red lines).

## Musical stimuli

We selected a performance by coauthor Elaine Chew of Ludwig van Beethoven's 32 Variations in C minor, WoO 80, recorded in one take without interruption or edits on a reproducing piano. The performance was then split into 33 distinct segments: the theme (named Tema, in Italian, is used throughout this paper), and the 32 variations on that theme. The music was chosen because the theme and variations, which are similar in musical style and ideas, cover a range of tempo and dynamics.

The theme is composed in the key of C minor. All variations share the same tonic, but the major mode is used from Var XII through Var XVI. The theme and most variations maintain an 8-bar structure with a 3/4 time signature, and a descending baseline, which are typical characteristics of the Chaconne [22]. In addition, rhythmic accents on the weak beats of the main melody are reminiscent of the French Sarabande [23, p. 750]. Var XXXII marks an exception to this pattern; it functions as an extended epilogue with a coda, notably departing from the main structure in its duration, number of bars, and harmonic progression.

A recurring phrase of ascending and descending notes is a prominent figure throughout the theme and variation set. A similar form of this figure (not necessarily preserving the same intervals) is reiterated four times in most variations with increasing stress, usually co-occurring with a rising pitch, followed by a slight pause and a sudden accent (*sforzando*), and ending with a slower conclusion, with decreased intensity, marked harmonically with a cadence to the tonic. See the Supplementary Materials to access: the entire score of the set (S1 File) and screen capture videos of individual performances with concurrent MIDI notes and audio waveforms–as visualized in CosmoNote–with the recorded performance (S1 Video).

## The study

There were five main stages to the study. Participants first completed a short questionnaire about their musical abilities (self-reported). Next, they calibrated their audio levels and did a hearing environment test. Then, they familiarized themselves with the interface using the Training Collection in CosmoNote, while guided by the experimenter. The main task consisted of marking the boundaries heard in the music and indicating the strength of each

boundary. Boundaries were defined as: "time points that separate a music stream into segments representing meaningful chunks of music, e.g., a musical idea or a musical thought. Boundaries not only separate a larger piece of music into smaller, coherent units, they also help listeners make sense of the music." Participants could choose between four strength levels, defined from 1 (weakest) to 4 (strongest). The full instructions are part of the supplementary materials (see S2 File). After completing the annotations, they were asked to complete a feedback survey. The study was designed to be completed in approximately 100 minutes.

## Study duration

Participants took different amounts of time to complete the study. The study duration was tracked using the first and last annotation timestamps, recorded by collection. We did not record each participant's time completing the questionnaires or annotating the Training collection. The study's mean duration was 56 minutes, with a standard deviation of 24 minutes. The median duration was 53 minutes; the shortest time was 10.5 minutes, and the longest was 2 hours.

## Study conditions

Participants were presented with different conditions combining two visual information layers, a waveform or a piano roll (notes without pedal data) and the audio information (the actual sound of the music). Each piece was presented in a fixed condition, preventing participants from changing the visualization/audio options in CosmoNote. During playback, an auditory cue was played by default at a boundary's timestamp for conditions including audio. However, participants were able to toggle it on or off. Three global study conditions were designed, from which seven total conditions were extracted, as seen in Fig 2 and described below:

i. **Audio and visuals:** Cross-modal conditions where both audio and visual information is presented. Participants were asked to listen to the music to annotate and were told they might use the visual representations to help them annotate. Three conditions fit these criteria.
   1: APW → Audio + Piano roll + Waveform
   2: AP → Audio + Piano roll
   3: AW → Audio + Waveform

ii. **Only visuals:** Unimodal conditions where only the audio information is not presented. Participants were reminded that they would see the representations but not hear the sound and asked to annotate visually. Three conditions fit these criteria.

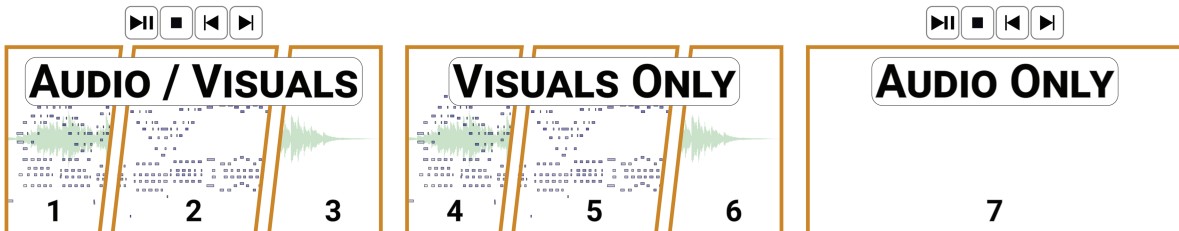

**Fig 2. Audio/Visuals study conditions.** Simplified interface representation of the seven study conditions. 1: APW, 2: AP, 3: AW, 4: PW, 5: P, 6: W, 7: A. Audio controls only show when the music can be played. Subdivisions are highlighted in orange.

4: PW → Piano roll + Waveform

5: P → Piano roll

6: W → Waveform

 iii. **Audio only:** Unimodal condition where only the audio information is presented; the visual information is absent. Participants were asked to listen to the music to annotate and were instructed that they could not see any music representations. One condition fits the criteria.

7: A → Audio

Each of the seven conditions was assigned to the pieces as a Latin Square. The order of the pieces participants annotated was shuffled. Thus, participants were divided into seven groups of eight people so that all the pieces would be annotated by a group in each of the seven conditions. All participants were exposed to all the conditions for the same number of pieces. Additionally, since experimental conditions are derived from the same musical source, there is no conflicting information between conditions; participants are only presented with more or fewer components. The PW visuals for all stimuli (Piano roll and Waveform overlapped) are shown the supplementary materials, as displayed in CosmoNote during the study.

## Grouping experimental conditions

To highlight the contrast between annotations placed using audio/visuals, we combined the seven experimental conditions detailed under study conditions, i.e., we pooled all boundary annotations from each condition, into two groups with common elements, then split into subgroups: (1) Conditions with audio, with two subgroups, labeled 'AV' and 'A'; and, (2) Conditions without audio, with three subgroups, labeled 'V', 'P', and 'W'. Annotations of each subgroup within 'Conditions with audio' were compared to annotations of each subgroup within 'Conditions without audio'. We do not focus on comparisons between visuals only because they do not provide helpful information for the central question of whether visuals or audio is prevalent. Thus, we choose 7 out of 10 possible combinations, as shown in Figs 3 and 4.

It is essential to remember that when grouping and comparing annotations with repeating conditions (e.g., APW vs. PW), we are not copying the same annotations into different groups because different people annotated each condition. We are comparing the support used to create each group of annotations. For example, AV contains different sets of annotations than V and A individually, even though they share experimental conditions.

We are particularly interested in comparisons 1, 2, and 7 in Fig 4. Comparisons 1 and 7 evaluate how close cross-modal annotations (visual/aural) are to unimodal annotations (visual or aural). Comparison 2 evaluates how close to each other are the two unimodal annotations (visual and aural).

## Removing outliers

The annotation data contained, among other information, the number of boundaries participants placed for each level by condition and piece. This number allowed us to calculate statistical descriptors such as the mean and standard deviation of the number of boundaries placed by piece. We identified, for example, that some participants marked an atypical amount of boundaries compared to the mean. Outliers were defined by boundary level as the number of boundaries that fell outside the upper outer fence, that is, more than three times the interquartile range (IQR) above the upper quartile. These annotations may be caused by participants not using all four levels (all annotations were done using only one level) or not understanding/respecting the annotation instructions. If data for a participant/level pair

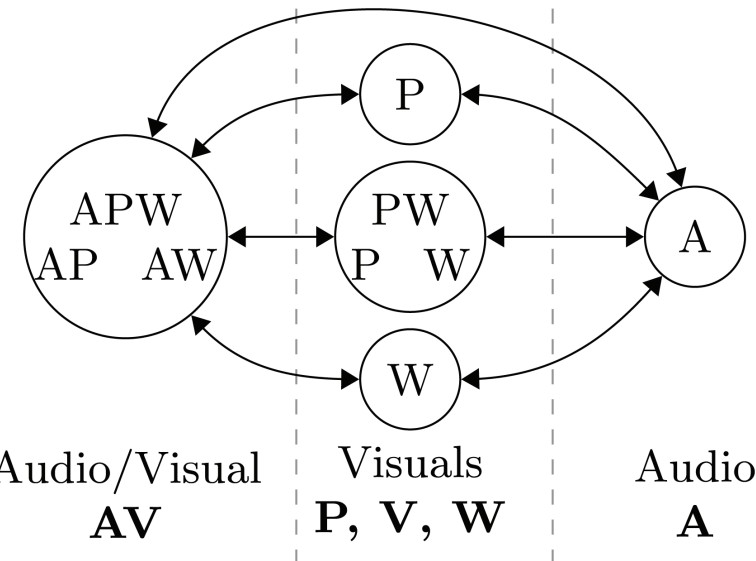

**Fig 3. Comparison scheme.** Conditions with audio (AV, A) are shown on the left and right columns. Conditions without audio (P, V, W) are shown on the center column.

| | V {PW, P, W} | P | W | A |
|---|---|---|---|---|
| **AV** {APW, AP, AW} | ① AV vs. V | ③ AV vs. P | ⑤ AV vs. W | ⑦ AV vs. A |
| **A** | ② A vs. V | ④ A vs. P | ⑥ A vs. W | |

**Fig 4. Comparison table.** Grouped study conditions are compared across rows and columns. Each cell contains a circled number indicating a specific comparison, ranging from 1 to 7.

were considered outliers, all boundary data for that level were removed from the participant's dataset. This data can be accessed at S1 Dataset.

## Distance metrics

This study analyzes differences between boundary annotations in performed music [18], aggregating and comparing the experimental conditions described above. In this context, the aggregate tendencies in the distribution of boundaries are considered representative of the segmentation emerging from a given study condition group, we therefore chose not to rely on ground truth annotations. This section defines the techniques used for the analysis of boundary annotations.

### F-measure

The weighted F-measure [19,21], abbreviated here as $F_\alpha$, measures how close estimated boundary annotations (within a specific time window) are to reference boundary annotations.

Since the F-measure is a similarity metric, we use $1 - F_\alpha$ as a distance between the boundary distributions.

**F-score.** The F-score, also called F-measure or $F_1$–score, is a common technique used for comparing flat (as opposed to nested) boundary annotations on a scale from 0 to 1 [19,21]. It measures how close estimated boundary annotations (within a specific time window) are to reference boundary annotations. Annotations that fall within the window are called hits or positive results. F-scores are computed using two parameters: Precision ($P$) and Recall ($R$). Precision measures positive results compared to the number of estimated boundaries, and recall measure positive results compared to the reference boundaries. Nieto et al. [21] introduced a variation accounting for perceptive bias favoring precision over recall using an $\alpha$ parameter smaller than 1 (we keep the default value $\alpha = 0.58$). Eq 1 recalls the expression of $F_1$ and $F_\alpha$:

$$F_1 = 2\frac{P \cdot R}{P + R} \qquad F_\alpha = (1 + \alpha^2)\frac{P \cdot R}{\alpha^2 P + R}. \tag{1}$$

To apply this technique to our data, we use a weighted variation of the $F_\alpha$–score, computing precision and recall values separately by strength levels (1–4), then combining them using a weighted sum, with scaled weights ranging from 0.25 to 1, and finally computing the $F_\alpha$–score using the weighted versions of precision and recall.

The F-score metrics were designed for situations where ground truth annotations exist. They have the advantage of providing values for precision and recall but do not give insight into the importance of individual boundaries in each annotation set. What is more, although a weighted variation of the technique is possible, as explained above, it fails to capture the interactions between boundaries of different strength levels individually. For example, if participants marked the same timestamp but used different strength levels, the $F_\alpha$–score does not consider such corresponding boundaries a hit. We argue that the unbalanced optimal transport distance solves these issues.

## Unbalanced optimal transport distance

The unbalanced optimal transport distance [24,25], abbreviated throughout this article as uOT, calculates a horizontal comparison and gives insight into which individual boundaries make profiles more distant by introducing a cost element to the objective function that allows us to remove individual boundaries, one by one, to minimize the distance between the two samples. To understand how this technique works we first define the optimal transport distance, as follows:

**Optimal transport.** Let $f$ and $g$ be two discrete functions from $\mathbb{N}$ with values in $\mathbb{R}^+$, representing two sets of weighted boundary annotations. The distribution function $F$ of $f$, and $\|f\|_1$, the total sum of the boundaries of $f$, are defined in Eq 2 as follows:

$$F(n) = \sum_{k=0}^{n} f(k), \qquad \|f\|_1 = \sum_{k=0}^{\infty} f(k). \tag{2}$$

Here, $n$ represents the upper limit of the summation for $F(n)$, which accumulates the values of $f(k)$ from $k = 0$ to $k = n$, while $\|f\|_1$ denotes the total sum of $f(k)$ over all indices $k \in \mathbb{N}$. Therefore, the distance associated with discrete optimal transport, $d_{OT}$ [26], between $f$ and $g$ is simply written in Eq 3 as:

$$d_{OT}(f,g) = \sum_{n=0}^{\infty} \left| \frac{F(n)}{\|f\|_1} - \frac{G(n)}{\|g\|_1} \right|, \tag{3}$$

where the factors $1/\|f\|_1$ and $1/\|g\|_1$ must be applied because the boundary sums of $f$ and $g$ are different. This distance can be interpreted as the cost of moving boundaries from $f$ to $g$, which is why it is also called the 'Earth mover's distance' [24]. The value of the optimal transport distance is usually proportional to the size of the area under both curves, meaning longer pieces yield larger distances. The lowest possible value of the optimal transport distance is 0 and even if the theoretical maximum value is infinite (because it depends on the piece's duration, which can also be infinite), the optimal transport distance calculates the proportion of changes to transform $f$ into $g$. To be able to compare the optimal transport distance of pieces of different lengths, we need to scale the distance by the duration of each piece.

In some cases, it might be preferable not to move a boundary from $f$ to $g$. For example, depending on the stimulus and the participant, a boundary may occur in $f$ at time $l$ without having a boundary in $g$ close to time $l$. In this case, it is preferable to remove such boundary from $f$ rather than move it towards $g$. To do this, we define $f_l$ as the $f$ function whose $l$ value is 0, see Eq 4.

$$f_l(k) = \begin{cases} 0 & \text{if } k = l, \\ f(k) & \text{otherwise.} \end{cases} \tag{4}$$

Additionally, isolated boundaries in one curve may distort the distance between the two. For example, an isolated boundary near the end of the curve $f$ must be moved towards at least one boundary of the curve $g$, which would inflate the optimal transport distance between the curves even if there is only one boundary that is different between them.

Allowing boundaries to be removed from $f$ or $g$ is similar to applying unbalanced optimal transport [25].

**Unbalanced optimal transport.** We compute the unbalanced optimal transport distance uOT between two weighted sets of boundaries by successively removing boundaries with a cost $c \in \mathbb{R}^+$ while it decreases the deformations required to turn one set of boundaries into another. In other words, if $l = \text{argmin}_{l \in \mathbb{N}}(d_{OT}(f_l, g))$ is the boundary that if removed minimizes the deformations required between the $f$ and $g$ boundaries, and that $d_{OT}(f_l, g) + cf(l) < d_{OT}(f, g)$, then $f$ is initialized to $f_l$ and we continue to remove boundaries. At the end, we have removed the set of boundaries $A$ to $f$ and $B$ to $g$ and to obtain the distance based on the unbalanced optimal transport between $f$ and $g$ in Eq 5:

$$d_{UOT}(f, g) = d_{OT}(f_A, g_B) + c \sum_{k \in A} f(k) + c \sum_{k \in B} g(k), \tag{5}$$

where $c$ is the cost of destroying (removing) boundaries, and $f_A$ is the function $f$ which is $f(k) = 0$ if $k \in A$. Thus, the cost determines the likelihood of removing a boundary in one of the two distributions. Higher costs discourage boundary removal because the distance between the distributions with the cost of removing such boundaries will be higher than between the original distributions. If the cost is too high, the unbalanced optimal transport distance will equal the optimal transport distance. Conversely, if it is too low, the algorithm will remove boundaries until both distributions have the same number of non-zero boundaries. We tested the method with different costs through iterative exploration and chose a cost $c = 2$, that yielded meaningful results, capturing subtle differences between distributions without removing too many boundaries.

The unbalanced optimal transport technique reveals which specific boundaries cause the largest discrepancies between any two curves $f$ and $g$. This insight represents a powerful tool for the analysis and interpretation of results. As mentioned above, distances computed with

this technique are scaled to be comparable between pieces of different durations. The code used to compute these distances can be accessed at S1 Code.

## Comparing between distance metrics

Fig 5 shows, at a glance, box plots used to compare the $1 - F_\alpha$ and uOT distance metrics. Each box plot shows a distribution with 33 points (all pieces) computed from a specific distance metric (panels) on one of the seven study conditions listed in Fig 4 (rows). The left panel displays the $F_\alpha$–score complement while the right panel displays the unbalanced optimal transport distance. Distances in individual panels are displayed from lowest to highest across the $x$-axis. The $y$-axis is divided into four rows. The first three rows in every panel contain two box plots. They compare either audio/visual (AV) or audio (A) annotations to one condition with visuals only (V, P, W). The last row contains only one box plot. It compares audio/visual (AV) to audio (A) annotations.

The same tendency is observed on the three top rows of all panels. First, compared to solely visual annotations, distributions of audio/visual conditions (AV) have a lower median distance than distributions where only audio was presented (A). Second, box plots between panels are visually similar, suggesting that both metrics generate comparable measured outcomes when analyzing boundary annotations. In other words, rows in Fig 5 show that hearing the music *and* looking at the visuals (AV) results in annotations more similar to those placed solely with visuals (V, P, W) than solely with audio (A). Access to the notes and the waveform simultaneously (V) enhances this similarity, compared to having only one visual at a time (P or W). We break down these relationships in more depth in the following section.

## Results

This section compares uOT distances between boundaries from the seven study conditions in Fig 4, that pooled annotations with/without audio information. We first examine the results

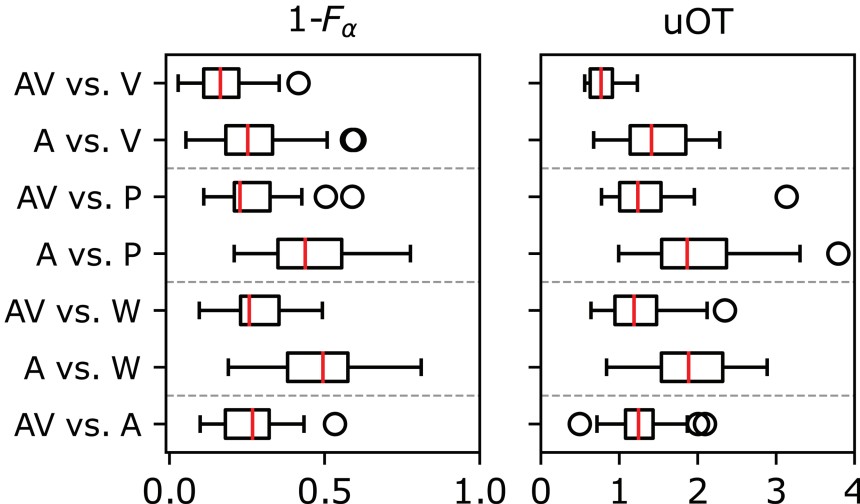

**Fig 5. uOT compared to $1 - F_\alpha$.** Box plots of the $F_\alpha$–score's complement (left panel) and the unbalanced optimal transport distance (right panel). The distances were computed on all 33 pieces in the WoO 80 set. A distribution of distances comparing grouped study conditions (aural/visual) are shown on each row.

by computing the mean and median distances from the WoO 80 set, providing a global ranking in which we see that V is closer to AV (i.e., the first position on the ranking) most of the time. We then inspect the results by looking at a piece-by-piece ranking of the comparisons and focusing on specific examples and outliers of the smallest and largest uOT distances. The descriptions mentioning musical pieces are focused primarily in the context of their segmentation structure and the presence or absence of visual and aural cues. Consequently, musical analysis of the score and performance are invoked only when needed.

Annotation profiles closest to AV are the gold standard for cross-modal comparisons, while annotation profiles closest to A are the gold standard for unimodal comparisons. Because our study collected boundary annotations in conditions that progressively presented participants with less information, we expected the best annotation results when participants had the most information layers. AV and A represent real-world prototypical structures and provide the most information in their categories. Thus, uOT distances closest to AV (in cross-modal) and closest to A (in unimodal) are better results.

## Global results

We computed distances for seven comparisons over 33 pieces of the set summarized in Table 1. Calculating the mean and median of these uOT distances produces rankings (based on the median distance) of grouped conditions. The smallest distance in Table 1 (where annotation profiles are the most similar) occupies the first position of the ranking while the largest distance (where annotation profiles are the most different) occupies the last position. For context, the global mean of uOT distances is 1.43 units with a standard deviation of 0.58 units; see Table 2.

Our results show that annotations with both aural and visual (cross-modal) components (AV) systematically rank above those with only one component (V, W, P, or A). Indeed, AV vs. V is first on the ranking in Table 1 meaning AV annotation profiles share the most attributes with V annotation profiles. Conversely, the last positions on the ranking suggest A annotation profiles share the least attributes with P or W annotation profiles.

The ranking positions on Table 1 are spread as per-piece rankings of uOT distances to form the heat map representation in Fig 6. Darker colors represent smaller distances while lighter colors represent larger distances. Rankings by piece allow us to identify outliers for closer

**Table 1. Ranking of uOT distance median and mean values by comparison.**

| Ranking | Comparisons | uOT distance | |
|---|---|---|---|
| | | median | mean |
| 1 | AV vs. V | 0.77 | (0.79) |
| 2 | AV vs. W | 1.19 | (1.27) |
| 3 | AV vs. P | 1.24 | (1.33) |
| 4 | AV vs. A | 1.25 | (1.27) |
| 5 | A vs. V | 1.41 | (1.47) |
| 6 | A vs. P | 1.87 | (1.97) |
| 7 | A vs. W | 1.88 | (1.91) |

Ranking positions are based on the median (third column) of uOT distance values.

**Table 2. Descriptive statistics for the ensemble of all uOT distance results.**

| Descriptor | mean | std | min | 25% | median | 75% | max |
|---|---|---|---|---|---|---|---|
| uOT distance | 1.43 | 0.58 | 0.50 | 1.00 | 1.29 | 1.77 | 3.79 |

| Comparison / Piece | 0 | 1 | 2 | 3 | 4 | 5 | 6 | 7 | 8 | 9 | 10 | 11 | 12 | 13 | 14 | 15 | 16 | 17 | 18 | 19 | 20 | 21 | 22 | 23 | 24 | 25 | 26 | 27 | 28 | 29 | 30 | 31 | 32 |
|---|---|---|---|---|---|---|---|---|---|---|---|---|---|---|---|---|---|---|---|---|---|---|---|---|---|---|---|---|---|---|---|---|---|
| AV vs. V | 1 | 1 | 1 | 1 | 1 | 1 | 1 | 1 | 1 | 1 | 1 | 2 | 1 | 1 | 1 | 1 | 2 | 1 | 1 | 1 | 1 | 2 | 1 | 3 | 1 | 1 | 1 | 3 | 1 | 1 | 1 | 1 | 3 |
| AV vs. W | 2 | 4 | 2 | 3 | 2 | 4 | 5 | 2 | 2 | 2 | 6 | 1 | 3 | 2 | 2 | 6 | 2 | 6 | 2 | 2 | 3 | 2 | 2 | 6 | 3 | 2 | 5 | 6 | 2 | 5 | 4 | 7 | 4 |
| AV vs. P | 6 | 3 | 3 | 2 | 4 | 6 | 3 | 3 | 5 | 3 | 2 | 5 | 6 | 3 | 6 | 4 | 3 | 2 | 3 | 6 | 1 | 3 | 5 | 4 | 2 | 5 | 7 | 2 | 5 | 2 | 2 | 2 | 5 |
| AV vs. A | 3 | 2 | 5 | 5 | 3 | 2 | 2 | 4 | 4 | 4 | 4 | 3 | 5 | 5 | 3 | 1 | 4 | 5 | 4 | 3 | 4 | 4 | 1 | 3 | 4 | 4 | 2 | 3 | 3 | 3 | 3 | 3 | 1 |
| A vs. V | 4 | 5 | 6 | 4 | 5 | 3 | 4 | 5 | 3 | 5 | 3 | 4 | 2 | 4 | 5 | 3 | 5 | 4 | 5 | 4 | 5 | 6 | 4 | 2 | 5 | 3 | 1 | 5 | 4 | 4 | 5 | 4 | 2 |
| A vs. P | 7 | 6 | 7 | 7 | 7 | 7 | 6 | 6 | 6 | 7 | 5 | 7 | 7 | 6 | 7 | 7 | 7 | 3 | 7 | 7 | 6 | 5 | 6 | 7 | 6 | 6 | 6 | 4 | 6 | 6 | 7 | 5 | 6 |
| A vs. W | 5 | 7 | 4 | 6 | 6 | 5 | 7 | 7 | 7 | 6 | 7 | 6 | 4 | 7 | 4 | 5 | 6 | 7 | 6 | 5 | 7 | 7 | 7 | 5 | 7 | 7 | 4 | 7 | 7 | 7 | 6 | 6 | 7 |

**Fig 6. uOT distance global rankings.** Heat map ranking uOT distances for all grouped comparisons (rows) and all 33 pieces in the WoO 80 set (columns).

inspection. We will later examine specific cases, analyzing noteworthy patterns for the set's smallest and largest uOT distances.

To depict uOT distances between each comparison on a piece-by-piece level, we created a two-dimensional representation, shown in Fig 7. We focus on the relative distance of points in the two-dimensional space rather than on the individual position of specific pieces. However, indicative labels show pieces numbered from 0 (Tema) to 32 (Var XXXII). Fig 7a shows all pieces and grouped conditions, where outliers of visual conditions P (purple) and W (green) are most evident. Two clusters, AV and V, are hidden in the first panel but highlighted at the center of Fig 7b, 7c, and 7d, which show only two conditions at a time to illustrate how they compare spatially. Fig 7b and 7d depict how visual/aural annotations (in red) are closer to solely visual (blue in Fig 7b) than solely aural (orange in Fig 7d) annotations. Visual information provides cues that auditory information lacks, making pieces cluster during MDS (we will explore this claim by describing examples in the next paragraphs). The same contrast can be seen in Fig 7b and 7c, which show pieces annotated visually clustering towards the center while those annotated with audio only are sparsely distributed around it.

Comparisons between conditions exemplified in Fig 8, and 9 examine how boundary annotations differ within a given piece. For visualization purposes, we employed Gaussian KDE profiles [28], on continuous boundary annotation curves, because of the convenience of using this technique for aggregating annotations and illustrating localized and global trends on the original data. Vertical lines are reserved exclusively to show removed boundaries, to avoid confusing them with other annotations. It is easier to see the similarity between superimposed profiles when depicted as aggregated curves. In all subsequent figures of this type, KDE boundary profiles of conditions with audio (AV and A) are drawn as solid lines, while shaded areas show boundary profiles of visual conditions (V, P, and W). Vertical dashed lines scaled to the height of the $y$-axis mark the boundaries removed by the uOT distance computation; they emphasize the timestamps where the two overlaid profiles differ the most. The rest of this article highlights specific pieces of the WoO 80 set. These examples illustrate cases where placing annotations with or without aural cues leads to clearly different boundary profiles. They also showcase interesting outliers where uOT distances are ranked differently than in Table 1. However, a compendium of interactive figures for all the pieces in the study, displaying the seven comparisons simultaneously, can be consulted online at S1 Fig.

Three pieces (Tema, Var XII, and Var XXXII) are plotted in Fig 8 to highlight positions 1, 5, and 7 of the median uOT distance ranking in Table 1. The similarity in profiles between AV (red line) and V (blue area) is evident in how the contours of their boundary profiles concur compared to other profiles; see Fig 8a and 8b. Even though the largest peaks are gathered around identical timestamps, subtle variations in the position of peaks indicate that distinct

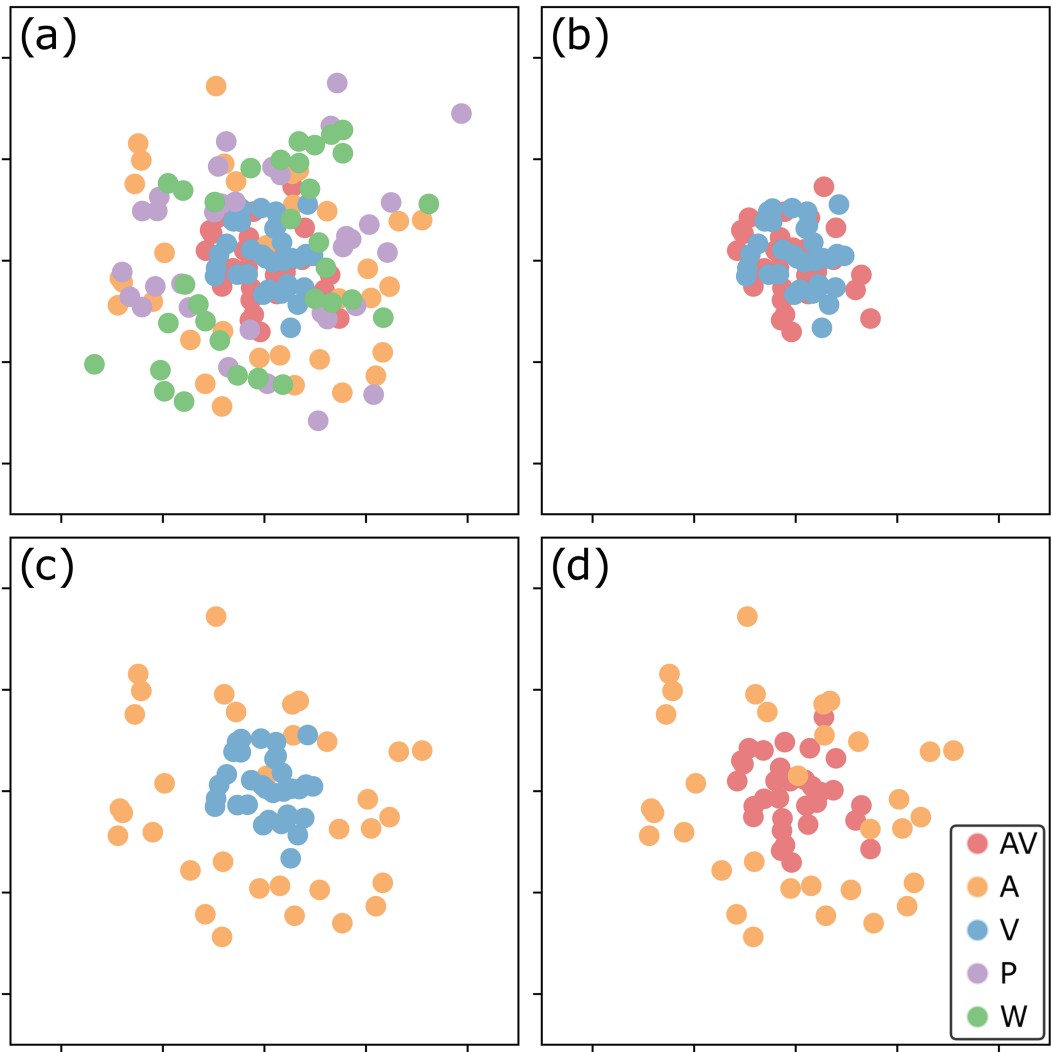

**Fig 7. MDS Representation of uOT Distances.** The pairwise uOT distances between grouped conditions (colored legend) are mapped into a two-dimensional space to visualize their relationship using multidimensional scaling (MDS) [27, p. 449]. Each number represents a piece. Each panel shows the same data, filtered by how many conditions are shown: (a) All conditions. (b) Audio/Visual vs. Visual. (c) Audio vs. Visual. (d) Audio/Visual vs. Audio.

conditions (that may vary per piece) give the most information about the segmentation structure of a piece. The overall distance between two conditions increases as differences in the position of peaks increase. Fig 8c contrasts the first (AV vs. V) and last (A vs. W) conditions of the ranking using an excerpt of Var XXXII. In more detail:

- Tema – Annotators attend to the right or left hand depending on the condition: Fig 8a shows the largest segmentation peak in profiles for all conditions, the *sforzando* at 10.5 s. The pause at 9.1 s, the change in dynamics to *piano* at 13.4 s, and the first quintuplet at 3.5 s are also worth noting. The boundaries align with the chords on the left hand (the starts of the black lines in the piano roll visuals) and the quintuplets (ornaments). Segments annotated aurally shift the distribution peaks earlier in time. For instance,

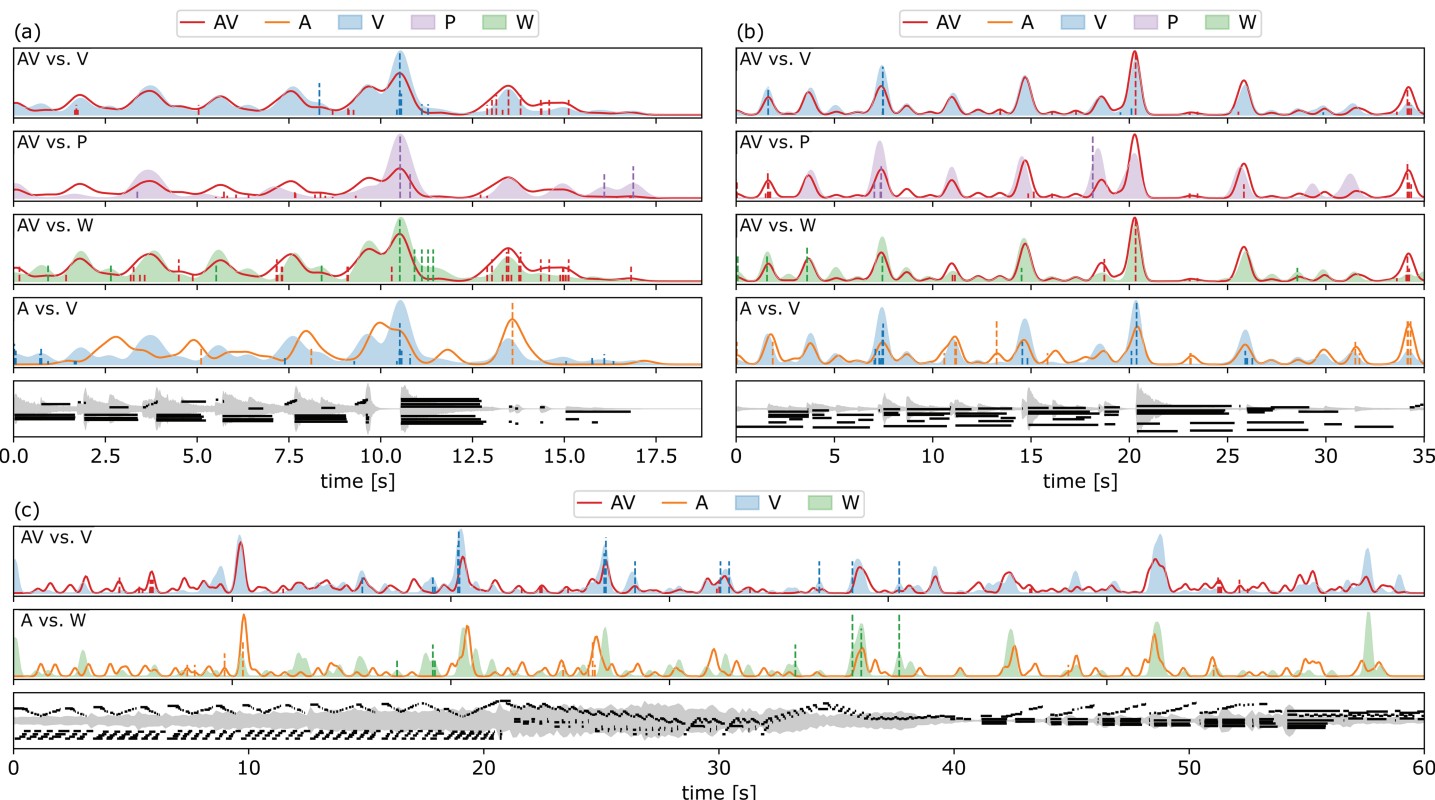

**Fig 8. Boundary profile comparisons.** Boundary annotations for the (a) Tema, (b) Var XII, and (c) an excerpt of Var XXXII of Beethoven's 32 Variations in C minor. KDE boundary profiles are drawn as either solid lines (conditions with audio) or shaded areas (conditions without audio). Boundaries removed by the uOT distance are shown as vertical dashed lines. The *y*-axis indicates the concentration of boundaries at each location, scaled to the highest peaks. Bottom panels of each sub-figure display waveform and piano roll visuals.

listeners are focused on melody notes (on the right hand) and, lacking a visual cue, the *sforzando* is marked early.

- Var XII – Annotators are attending to the starts of repeated patterns: Boundary profiles in Fig 8b appear roughly similar for all conditions. As before, the pause at 19.5 s and *sforzando* at 20.3 s are the most salient characteristics of the profiles annotated with the music (AV and A, the red and orange lines). Visual annotations primarily (V, P, W), but also audio/visual annotations (AV), assign more importance to the accent on the reiteration of the main motive at 7.5 s than audio annotations (A). As before, segmentation is driven by the first notes of the arpeggiated chords on the left hand (the longest black lines at the bottom of the piano roll visuals).

- Var XXXII – Waveforms are less helpful than audio in longer sections with subtler amplitude changes: Fig 8c zooms in a one-minute excerpt showing three sections of this piece. First, comparing between visual annotations we see that many peaks that appear in the shaded blue area (V) are absent from the shaded green area (W), indicating that they come from the second component in V, the piano roll (P). The section change at 20.7 s is less evident with only the waveform representation (green) than with the piano roll (see black lines at the bottom of the figure). This could be explained, for example, by the abundance of notes and overall *crescendo* making this change difficult to see in the waveform. Second, the pause and dynamic change to *pianissimo* at 42.7 s are evident in

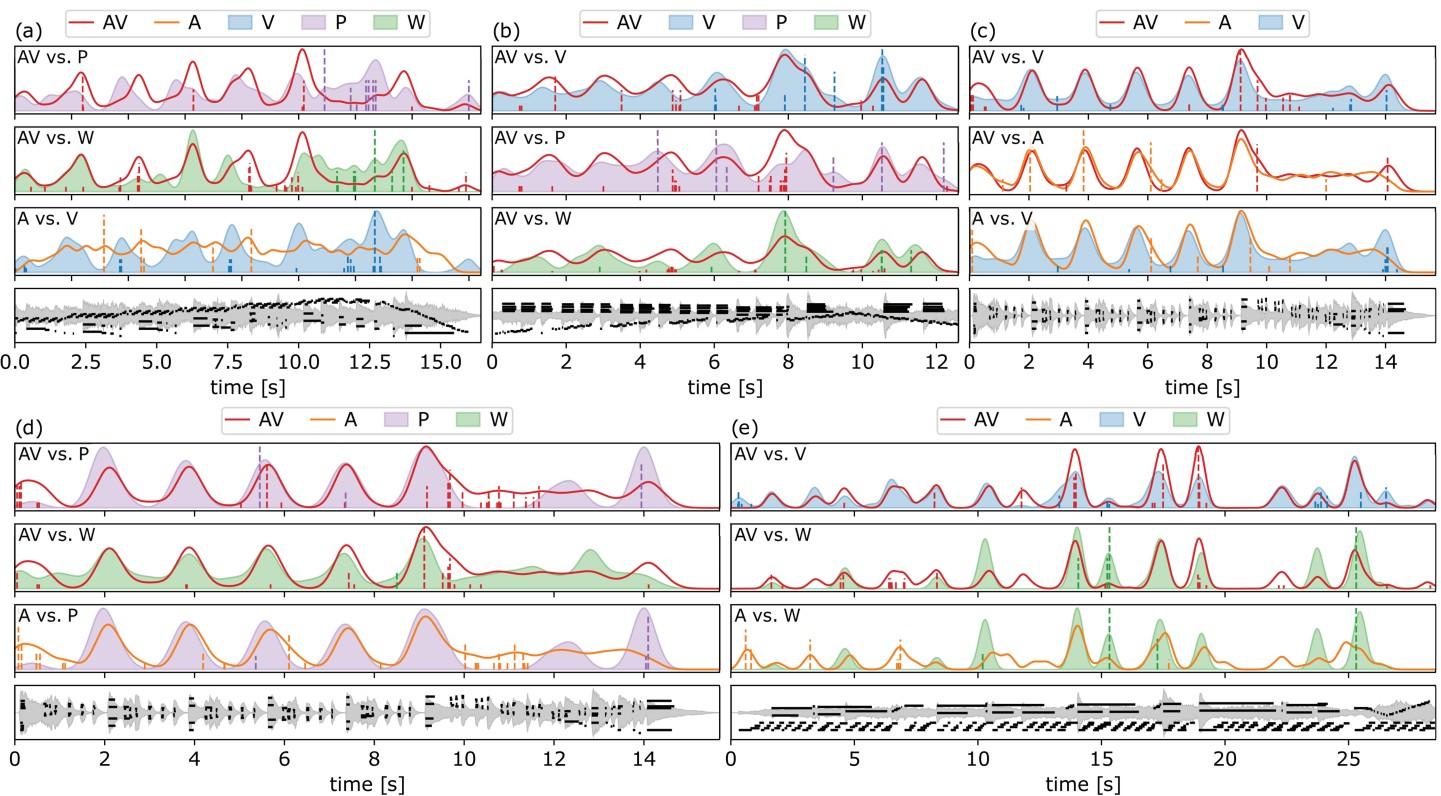

**Fig 9. Boundary profile comparisons (smallest/largest distances).** Boundary annotation profiles for (a) Var XI, (b) Var XX, (c) Var XXVI (smallest uOT), (d) Var XXVI (largest uOT), and (e) Var XXXI of Beethoven's 32 Variations in C minor. KDE boundary profiles are drawn as either solid lines (conditions with audio) or shaded areas (conditions without audio). Boundaries removed by the uOT distance are shown as vertical dashed lines. The $y$-axis indicates the concentration of boundaries at each location, scaled to the highest peaks. Bottom panels of each sub-figure display waveform and piano roll visuals.

all representations. However, annotations with audio shift the placement of the boundary after the first C major chord, to the B3 eighth note at 42.1 s. Third, even though the *sforzando* at 54.2 s on the left hand was visually more salient, annotations with audio placed more boundaries on the harmonic motive on the left hand at 52.7 s (shown by a peak on the orange curve, A) that concludes with the *sforzando* in question.

## Smallest distances

Smallest distances represent results where the annotation curves between two comparisons are the most similar. For this ranking, we start by considering all pieces and comparisons equally. That is, 231 points from 33 pieces and 7 comparisons. Then, we keep only the smallest distances in each piece (i.e., distances ranked 1 represented by the darkest colors on Fig 6) to see which pieces have a ranking that is different from the global ranking in Table 1. The smallest of all uOT distances is the comparison AV vs. A for Var XXXII, the set's longest piece. For this variation, the annotation profiles indicate that participants could obtain the most similar segmentation structure of the piece from cross-modal (visual and aural) or unimodal aural, instead unimodal visual cues (see S1 Table of the supplementary materials for specific values). For this piece, listening is almost as good as listening and seeing. A possible explanation is that participants annotating visually tended to place boundaries without using the zoom functionality, which may lead to larger discrepancies between timestamps in longer pieces such as

Var XXXII. Here, the auditory cues (including the boundary cue) would give participants a better anchor for placing boundaries on the interface.

Looking at the 33 uOT distance results ranked 1, piece by piece, the smallest uOT distances are between AV vs. V (27 pieces), AV vs. A (3 pieces), AV vs. W (1 piece), AV vs. P (1 piece), and A vs. V (1 piece). Here, one could argue that 32 out of 33 pieces are consistent with the results reported in Table 1. Fig 9a–9c highlights the subtlety of these three exceptions, where AV vs. V is still ranked second or third. In more detail:

- Var XI – Visually apparent note patterns can be misleading: Annotations made using the waveform only (W) noticeably resemble those placed with audio and visuals (AV) more than the other conditions Fig 9a. The red vertical dashed lines (removed boundaries in AV) over the green shaded area (denoting W) show that without hearing the music, annotators miss the accent at 4.3 s and place the accent at 8.2 s too early because the loudness changes in these accents are too fast and subtle to be salient on the waveform. The presence of the piano roll visualization (P) and the music (A) shifted the segmentation (as evidenced by the peaks on the solid red curve, AV) toward the right-hand patterns (jagged black lines in the piano roll visuals) at the bar level. Additionally, the note visuals enhance a beat the performer skipped at 12.6 s, which may have been visually mistaken for a pause (with a peak on the shaded purple area, P) but was aurally less striking in context, with the left hand continuing the descent.
- Var XX – Layering visuals reveals key details: As seen in Fig 9b with the green shaded area, the waveform visualization (W) favors the accented chords on the right hand. It is only with the addition of the piano roll visuals (P) that the segmentation on the melodic motifs of the left hand becomes more apparent to the participants.
- Var XXVI – Visuals bias segmentation toward the score structure: As with Var XII, all conditions have similar boundary profiles. Fig 9c shows that visual conditions (P, W, V) did not lead to segmentations capturing the performance. The performer chose to linger on the first notes of the gallop figures at each bar, creating a difference in segmentation that is only resolved by listening to the music. Both the waveform and the piano roll visuals privileged the accented chords with octaves on both hands at the bar level, seen as similar transients on the waveform and vertical lines on the piano roll.

## Largest distances

Largest distances represent results where the annotation curves between two comparisons are the least similar. As before, considering all pieces and all comparisons equally and keeping only the largest distances (i.e., all distances ranked 7 represented by the lightest colors on Fig 6). The largest of all uOT distances is the comparison A vs. P for Var XIV (see S2 Table of the supplementary materials for specific values). This result is consistent with Table 1. Annotations for this piece, placed with audio only compared to those with the piano roll only, present the largest differences among all annotations. Additionally, the result suggests that the visual pattern created by the notes only led to annotations that conflicted with annotations done with audio only (see Global Results).

When grouping the pieces before ranking comparisons, the largest uOT distances are shared between three comparisons: A vs. W (16 pieces), A vs. P (15 pieces), AV vs. P (1 piece), and AV vs. W (1 piece). Here, the tendency for 31 out of 33 pieces is expected (see Table 1), i.e., annotations placed only with audio are less similar to those placed only with one visual layer (either P or W). Two pieces, Var XXVI and Var XXXI, can be considered exceptions

although they are not too far from the global trend. For Var XXVI, A vs. P is the second-largest distance after AV vs. P, while for Var XXXI, A vs. W is the second-largest distance after AV vs. W. For these outliers in Fig 9d-e, annotations made from a particular visual layer (P or W) were closer to audio alone than to audio paired with visuals. More specifically:

- Var XXVI – The audio is best supported by well-defined transient peaks on the waveform: This time, Fig 9d focuses on comparing unimodal visual conditions (P and W) to cross-modal and unimodal aural conditions (AV and A). In this case, the piano roll visualization (P) favors two peaks (12.2 s and 14 s) over the waveform (W). As mentioned before, this segmentation is best understood when hearing the music. Accents during the descending motives between 10 s and 12 s were missed by listeners, as evidenced by the concentration of destroyed boundaries (dashed lines) in that zone. The transients on the waveform (W) convey more of the segmentation here than the piano roll (P), which is insufficient to emphasize prominence, making it rank last when compared to AV.
- Var XXXI – Timing cues such as syncopation are harder to notice on waveforms than through attentive listening: Differences between the green shaded area (W) and the red and orange solid lines (AV and A, respectively) in Fig 9e show that annotations created solely with the waveform visual lack boundaries marked when audio and the notes were present. We see that, for example, the syncopated sixteenth notes (at 3.3 s or 22.2 s) and the thirty-second-note quintuplet (at 6.3 s) are not noticeable with the waveform alone. Layering the piano roll visualization (P) with the waveform (W) makes these notes more salient, possibly competing with the accents that participants heard with the music alone, which are compatible with the more evident transients of the waveform.

## Discussion

Our results, compared to the full information standard (AV), show that visuals-only segmentations (V, P, W) are ranked above audio-only segmentations (A) in terms of closeness to annotations with full information. We believe visual information helps to create a coherent higher-level structure segmentation, while auditory information is best used to resolve specific contradictions by identifying the music's lower-level, subtle segmentation structures. Visuals provide a spatial and temporal logic that is indicative of large-scale structures (i.e., they create salient patterns), but rely on their color, shape, and layout to convey identifiable information [16]. These representations must then be used in service to the ear, not as a replacement for careful listening. Scientific music representations are purposefully conceived to emphasize certain aspects of the sound while neglecting others.

Visuals in our study (piano roll and waveform) contain only specific but incomplete information while audio contains broad but complete information. In other words, visuals simplify the audio source by reducing all its properties into a two-dimensional illustration. Piano roll representations show pitch, duration, and velocity; they may assist in quickly spotting visual patterns like groups of notes that form a chord, an ascending scale, or a repeated sequence, but do not provide cues as to the duration and intensity of the attack/release of notes. Waveform representations show amplitude and duration; they may help to estimate properties like transients, loudness, and pauses more accurately, but they lack information about specific notes. Audio stimuli are not only the source of these basic properties (pitch, duration, amplitude), but give access to additional properties such as the music's tempo, the instrument's timbre, and the room acoustics through hearing.

Visual layers thus accentuate crucial information and divert the annotator's attention from music features that may not be necessary for a specific analysis but constitute crucial components of a performance. In this sense, adding visuals to the audio gives people more support, a form of cognitive scaffolding for accurately identifying otherwise less salient patterns and marking a given boundary's start and end times. However, as expected, removing the audio component to leave only the visuals complicates the task by obscuring subtle acoustic cues, as reported by some participants. They described that annotating uniquely visual representations of the music made them feel lost, discouraged, and less confident about the cues they used to identify boundaries.

To insist on the advantages of adding visual layers for structure annotations, annotations created with more information are more accurate than those where less information was available. Gingras et al. [29] argue that performance enhancement in tasks with cross-modal stimuli is not due to simple target redundancy (two stimuli rather than one) but is likely due to increased error reduction. For instance, boundary placement after a pause is made more accurate by following visual cues that indicate the start of notes, which are missing in uni-modal audio conditions.

Going back to the results of the ranking in Table 1, the first position suggests a preeminence of combined visual stimulation over aural stimulation across pieces, consistent with the cognitive scaffolding paradigm and the work of Platz and Kopiez [9] and Tsay [8] on cross-modal stimulation in music perception. However, the second, third, and fourth positions in that ranking are more dependent on individual characteristics of pieces, i.e., there are some pieces where the annotations profiles are closer to the standard (AV) when participants focus on different visual components while annotation profiles are closer to AV in other pieces when visuals are removed altogether. How can we interpret this?

Results, where the distances AV vs. W are small are, on average, the second most similar comparison, followed by AV vs. P and AV vs. A, with individual fluctuations (see Figs 7d and 6). Indeed, some examples are presented in the Results to illustrate exceptions to these patterns. For certain pieces, such as Var XI, the waveform (W) contributed more relevant structural information than both visuals (V), the notes (P), or the audio (A). For other pieces, conflicts between audio and visuals (in the cross-modal condition) are resolved in favor of the audio. In any case, since AV vs. A is consistently ranked lower than AV vs. V, the assumption of visual over aural sensory preference is supported.

Some of the challenges we faced are inherent to the design choices of this study. We will discuss the generalization of our results, the role of musical abilities, and the choice of using only boundary annotation.

First, generalizing musical annotation results is complex because of the many unique characteristics of each stimulus. Instead of presenting various musical styles and practices, our study privileges the differences between modalities. It minimized the differences in stimuli, employing music with a consistent structural framework and short durations so that participants could annotate many pieces relatively quickly. Moreover, we counterbalanced the possibility of a perceived global structure emerging from the theme and variation form by presenting the 33 pieces in a shuffled order.

Second, we did not examine the differences between musician and non-musician participants because there is evidence that musical training has a limited effect on tasks involving the perceptual segmentation of music [30], although we recognize the influence of specialization and familiarity on the cognitive processing of segmentation structures [31] and music notation and representation. For example, Møller et al. [32] show how the specialization of

brain structures in professional musicians makes their segmentation approach different from non-musicians. Nevertheless, there is evidence that musicians improve less with the help of visual cues than non-musicians in objectively quantifiable tasks, such as pitch detection [32]. Participants with previous experience using waveform/piano roll representations may have had an advantage in our task over those without. Such an advantage is not exclusive to musicians, who would be ahead in working with music notation. Indeed, both groups may have expressed similar learned cross-modal correspondences (e.g., increase in pitch and height) when looking at the notes of the piano roll [33]. Ultimately, we chose to analyze annotations assembled from a balanced group with varying musical abilities. What is more, contributions of both musicians and non-musicians (with more participants potentially coming from the latter group) will be grouped in large-scale citizen science studies, which involves voluntary contributions of mostly non-experts. Even if non-experts outnumber experts, Visscher and Wiering provide evidence that non-expert annotations are suitable for evaluating segmentation boundaries in polyphonic music [34].

Third, we intentionally restricted CosmoNote's interface to allow only boundary placement and not other available structural annotations such as regions or groups, as mentioned above. This is because we wanted participants to focus on a straightforward task. Likewise, we allowed the use of labels, though we did not take them into account for our results. Future work is needed to fully understand how expressive prosodic structures are perceived and marked in performed music with broader annotation data such as labeled boundaries, regions, note groups, and comments.

Even if visual cues are sometimes overemphasized or potentially misleading, the use of combined visuals (waveform and piano roll) brings annotation profiles closest to AV, judged in our study as the gold standard. Thus, we ought to consider the biases that may be introduced in cross-modal stimuli when designing annotation campaigns. Having scientific visuals providing concrete information about the music (e.g., waveform and piano roll) is better than not to have them. However, we can then think about ways to tackle the obstacles of perceptual biases. For instance, in free annotation tasks, participants may deliberately choose to show or hide a visual component without being aware of the subtle changes that this action may produce in their annotations. To counterbalance this effect, we can take a few approaches. Beginner annotators may be shown specific visualizations in a defined order to minimize bias. Segmentation tasks can be started by a group annotating with restricted visual representations, and these annotations would then be given to a second group to refine without restricted visuals. Annotators' actions could also be tracked more precisely so that annotations created with different modalities are analyzed separately.

There are other visuals available in CosmoNote (e.g., loudness, tempo, harmonic tension) that have the potential to be compared against unimodal audio stimuli for segmentation purposes. Work by Guichaoua et al. [20] illustrates how loudness and tempo curves can be used to estimate the segmentation structure of performed music. Designing experiments to study such differences in cross-modal annotations with loudness and tempo data through our musical prosody annotation protocol could help determine the advantages and disadvantages of different visual inputs. For example, single curves (tempo only or loudness only) are more straightforward to read because they encode information in only one dimension but, at the same time, they limit the possibilities of interpreting the musical meaning to that dimension. We also need to be aware of the possibility of a detrimental increase in cognitive load when using too many visual cues simultaneously, which should be avoided for a better experience. Future studies may reveal the limits on the amount of helpful concurrent

cross-modal information participants are comfortable with for annotating structures in performed music.

## Conclusion

This article has focused on boundary annotations created using either unimodal (visual or aural) or cross-modal (visual/aural) stimuli. We have shown that visuals play a considerable role in shaping boundary annotations of a recorded performance. Our results show that unimodal visual annotations on their own provide a reasonable global structural segmentation as defined by high-level boundaries compared to cross-modal ones—annotations marked with visual only input (V) is closest to full aural-visual information annotations (AV). Unimodal aural annotations are crucial to clarify detailed structures as represented by lower-level boundaries. We thus emphasize the benefits of having a cross-modal stimulus by using scientific visual representations to get the most out of human annotated musical structures in large-scale data collection. For example, studies where non-professionals are involved, such as citizen science projects [17].

Finally, the use of the unbalanced optimal transport (uOT) distance provides detailed information about the relative importance of specific boundaries. It allowed us to (1) examine subtle differences in the segmentation of individual pieces (i.e., which samples differ the most between any two distributions); and, (2) rank these differences across the whole theme and variation set (i.e., which pieces favored visuals over audio and which pieces were exceptions to that tendency). We believe this technique is particularly well adapted to analyzing and assessing similarity between structural segmentations in music.

## Supporting information

**S1 Fig. Boundary profiles.** Interactive figures with complementary profile comparisons of unimodal vs cross-modal annotations for all musical stimuli available at the URL https://doi.org/10.6084/m9.figshare.27918021.

**S1 Dataset. Annotation data.** JSON file containing boundaries with strength levels and timestamps by participant ID and study condition available at the URL https://doi.org/10.6084/m9.figshare.27917934.

**S1 Code. GitHub repository.** The code used to compute the unbalanced transport (uOT) distances is available at the URL https://github.com/erc-cosmos/unbalanced-optimal-transport-distance and the URL https://doi.org/10.6084/m9.figshare.27917781.

**S1 File. Score.** The complete score of Beethoven's 32 variations in C minor can be found at the URL https://doi.org/10.6084/m9.figshare.24635568.

**S2 File. Annotation instructions.** Document with the instructions given to participants during the study.
(PDF)

**S1 Video. Performances.** Videos of individual performances visualized in CosmoNote (waveform and piano roll) can be found at the URL https://doi.org/10.6084/m9.figshare.c.6755340.v1.

**S1 Table. Smallest distances.** Table with smallest uOT distances between all comparisons.
(PDF)

**S2 Table. Largest distances.** Table with largest uOT distances between all comparisons. (PDF)

## Author contributions

**Conceptualization:** Daniel Bedoya, Paul Lascabettes, Elaine Chew.

**Data curation:** Daniel Bedoya, Lawrence Fyfe.

**Formal analysis:** Daniel Bedoya.

**Funding acquisition:** Daniel Bedoya, Elaine Chew.

**Investigation:** Daniel Bedoya.

**Methodology:** Daniel Bedoya, Lawrence Fyfe, Elaine Chew.

**Project administration:** Elaine Chew.

**Resources:** Daniel Bedoya.

**Software:** Daniel Bedoya, Paul Lascabettes, Lawrence Fyfe.

**Supervision:** Elaine Chew.

**Validation:** Daniel Bedoya, Elaine Chew.

**Visualization:** Daniel Bedoya.

**Writing – original draft:** Daniel Bedoya, Paul Lascabettes, Elaine Chew.

**Writing – review & editing:** Daniel Bedoya, Elaine Chew.

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
