## [Decision Letter · Decision Letter 0]

17 Mar 2025

PONE-D-24-54698Science of music-based citizen science: How seeing influences hearingPLOS ONE

Dear Dr. Bedoya,

Thank you for submitting your manuscript to PLOS ONE. After careful consideration, we feel that it has merit but does not fully meet PLOS ONE’s publication criteria as it currently stands. Therefore, we invite you to submit a revised version of the manuscript that addresses the points raised during the review process.

**Two experts in the field have carefully reviewed the manuscript entitled “Science of music-based citizen science: How seeing influences hearing “. You can find their comments below. They both had very positive comments on the manuscript but also one of tthem requested clarifications of some parts and a new figure 7. **

**In light of these reviews, I am requesting a minor revision and resubmission, in which you will need to respond to each point in the second review.**

We look forward to receiving your revised manuscript.

Kind regards,

Bruno Alejandro Mesz, Ph.D.

Academic Editor

PLOS ONE

Journal Requirements:

1. When submitting your revision, we need you to address these additional requirements. Please ensure that your manuscript meets PLOS ONE's style requirements, including those for file naming. The PLOS ONE style templates can be found at https://journals.plos.org/plosone/s/file?id=wjVg/PLOSOne_formatting_sample_main_body.pdf and https://journals.plos.org/plosone/s/file?id=ba62/PLOSOne_formatting_sample_title_authors_affiliations.pdf 2. Thank you for stating in your Funding Statement: This result is part of the project COSMOS that has received funding from the European Research Council under the European Union's Horizon 2020 research and innovation program (Grant agreement No. 788960). The experiments conducted at the INSEAD-Sorbonne Université Behavioral Lab were funded by the French Excellence Initiative (Idex) at the Sorbonne Université. Paul Lascabettes was supported by a Specific Doctoral Contract for Normaliens (CDSN).  Please provide an amended statement that declares *all* the funding or sources of support (whether external or internal to your organization) received during this study, as detailed online in our guide for authors at http://journals.plos.org/plosone/s/submit-now.  Please also include the statement “There was no additional external funding received for this study.” in your updated Funding Statement. Please include your amended Funding Statement within your cover letter. We will change the online submission form on your behalf. 3. Thank you for stating the following in the Acknowledgments Section of your manuscript: This result is part of the project COSMOS that has received funding from the European Research Council under the European Union’s Horizon 2020 research and innovation program (Grant agreement No. 788960). The experiments conducted at the INSEAD-Sorbonne Universit´e Behavioral Lab were funded by the French Excellence Initiative (Idex) at Sorbonne Universit´e. Paul Lascabettes was supported by a Specific Doctoral Contract for Normaliens (CDSN). We note that you have provided funding information that is not currently declared in your Funding Statement. However, funding information should not appear in the Acknowledgments section or other areas of your manuscript. We will only publish funding information present in the Funding Statement section of the online submission form. Please remove any funding-related text from the manuscript and let us know how you would like to update your Funding Statement. Currently, your Funding Statement reads as follows:  This result is part of the project COSMOS that has received funding from the European Research Council under the European Union's Horizon 2020 research and innovation program (Grant agreement No. 788960). The experiments conducted at the INSEAD-Sorbonne Université Behavioral Lab were funded by the French Excellence Initiative (Idex) at the Sorbonne Université. Paul Lascabettes was supported by a Specific Doctoral Contract for Normaliens (CDSN). Please include your amended statements within your cover letter; we will change the online submission form on your behalf. 4. When completing the data availability statement of the submission form, you indicated that you will make your data available on acceptance. We strongly recommend all authors decide on a data sharing plan before acceptance, as the process can be lengthy and hold up publication timelines. Please note that, though access restrictions are acceptable now, your entire data will need to be made freely accessible if your manuscript is accepted for publication. This policy applies to all data except where public deposition would breach compliance with the protocol approved by your research ethics board. If you are unable to adhere to our open data policy, please kindly revise your statement to explain your reasoning and we will seek the editor's input on an exemption. Please be assured that, once you have provided your new statement, the assessment of your exemption will not hold up the peer review process.

Reviewers' comments:

Reviewer's Responses to Questions

**Comments to the Author**

1. Is the manuscript technically sound, and do the data support the conclusions?

Reviewer #1: Yes

Reviewer #2: Yes

2. Has the statistical analysis been performed appropriately and rigorously? 

Reviewer #1: Yes

Reviewer #2: Yes

3. Have the authors made all data underlying the findings in their manuscript fully available?

Reviewer #1: Yes

Reviewer #2: Yes

4. Is the manuscript presented in an intelligible fashion and written in standard English?

Reviewer #1: Yes

Reviewer #2: Yes

5. Review Comments to the Author

**Reviewer #1:** In the present study, participants annotated music segment boundaries for Ludwig van Beethoven's 32 Variations in C minor, WoO 80 under unimodal (visual or auditory) or crossmodal (visual/aural) conditions. The annotations created web-based with CosmoNote were compared using boundary credence profiles and the optimal transport distance. The results show that visual inputs strengthen the global segmentation approach, while audio allows for smaller details and differentiation. Overall, audio and visual inputs facilitate annotations, with visual cues better capturing large structures and audio better capturing subtle nuances. However, a cross-modal stimulus has a greater overall advantage, especially in studies involving non-experts, such as citizen science projects.

Even though I may not fully have understood the mathematical derivation of the formulas for the boundary credence profiles and the optimal transport distance, the study fully convinced me with its clear research question, its clean and consistently applied method and its stringent discussion of the results obtained.

**Reviewer #2:** The article presents a rigorous investigation into cross-modal perception of music segments, demonstrating how visual representations (e.g., waveform or piano roll notations) can modify listeners' interpretation of musical structure within an auditory-only context. The proposed methodology is built around human-subject segmentation trials, and the statistical analysis of the results is conducted without a strict ground truth. Instead, the distribution over annotated boundaries is considered as a reference.

Overall, the text is clearly written and accessible, the "Distance metrics" section, however, was a little hard to follow. In particular the definition of uOT.

The article tackles a highly pertinent topic, and its contributions are presented throughout the text. However, fixing the following issues would significantly improve the manuscript:

- line 20: "such as going up in pitch and space" - please rephrase, "going up in space" sounds vague

- line 39: "eometrical" - typo

- Equation 2: what is k and n?

- line 246: Suggestion : "What is more, although a weighted variation of the technique is possible, as explained above, it fails

to capture the interactions between boundaries of different strength levels individually." (if -> although, strengths -> strength)

- line 267: "also called the ‘Earth mover’s distance’" (please add reference)

- line 276: Equation has no number (is this intentional?)

- line 286: "successively removing boundaries with a cost (c) while" - please specify c before using it on line 289

- line 287: "to to" typo

Figure 5: what are the 1-Falpha and uOT value ranges? Please add x-axis to the Figure.

Figure 7: Illegible. Please regenerate this image, there is no much to see in it. No legends, numbers are overlapping. Why MDS, and not PCA or TSNE?

Figure 8 and 9: Very good!

6. PLOS authors have the option to publish the peer review history of their article (what does this mean?). If published, this will include your full peer review and any attached files.

Reviewer #1: **Yes: **Claudia Bullerjahn

Reviewer #2: **Yes: **Rodrigo Borges

---

## [Author Response · Author response to Decision Letter 1]

30 Apr 2025

Dear reviewers,

Thank you for your time and for your comments regarding our article entitled “Science of music-based citizen science: How seeing influences hearing”.

The following paragraphs contain a description of the changes we made to the manuscript. Line numbers refer to the original version of the manuscript for comparison.

Form (typos, missing notation, etc.)

- Rephrased unclear text on line 20.

- Corrected typos on lines: 39, 287.

- Clarified indices k and n form the expressions in Equation 2:

- Accepted suggestions on line 246.

- Added a reference to support the Earth mover’s distance denomination.

- Included a number and a reference to the equation on line 276.

- Specified the constant c, at line 286.

- Added x-tick labels to the x-axis in Figure 5.

- Regenerated Figure 7. The numbers were removed, the dots were enlarged, their opacity increased, and the legend now contains a border to differentiate it from the points.

Content

Figure 5: What are the F_alpha and uOT value ranges?

The F-measure and its variants have a range of 0 to 1. The lowest possible value of the uOT distance is 0. However, its highest value depends on the length of the piece, as stated in the manuscript. Thus, exact values are difficult to provide.

Figure 7: Why MDS, and not PCA or TSNE?

PCA works best when multiple variables can be used to support the 2D projection. The principal components in the projection are those that best explain the variance in the data. We do not compare multiple variables in our dataset; thus, PCA is less useful in our study. While t-SNE is also a good option for dimensional scaling, it introduces unnecessary complexity (e.g., tuning parameters such as perplexity, method, and early exaggeration). In addition, this technique tends to prioritize the distance between neighbors. We chose MDS because it would preserve the distances between all the points in the distribution as much as possible while being simple to implement.

Sincerely,

---

## [Editor Report · Decision Letter 1]

6 May 2025

Science of music-based citizen science: How seeing influences hearing

PONE-D-24-54698R1

Dear Dr. Bedoya,

We’re pleased to inform you that your manuscript has been judged scientifically suitable for publication and will be formally accepted for publication once it meets all outstanding technical requirements.

Kind regards,

Bruno Alejandro Mesz, Ph.D.

Academic Editor

PLOS ONE
---

## [Editor Report · Acceptance letter]

PONE-D-24-54698R1

PLOS ONE

Dear Dr. Bedoya,

I'm pleased to inform you that your manuscript has been deemed suitable for publication in PLOS ONE. Congratulations! Your manuscript is now being handed over to our production team.

Kind regards,

on behalf of

Dr. Bruno Alejandro Mesz

Academic Editor

PLOS ONE